# Non-Covalent Bruton’s Tyrosine Kinase Inhibitors in the Treatment of Chronic Lymphocytic Leukemia

**DOI:** 10.3390/cancers15143648

**Published:** 2023-07-17

**Authors:** Skye Montoya, Meghan C. Thompson

**Affiliations:** 1Sylvester Cancer Center, University of Miami, Coral Gables, FL 33146, USA; 2Leukemia Service, Memorial Sloan Kettering Cancer Center, New York, NY 10021, USA

**Keywords:** chronic lymphocytic leukemia, Bruton’s tyrosine kinase, ibrutinib, acalabrutinib, zanubrutinib, pirtobrutinib, mechanisms of resistance

## Abstract

**Simple Summary:**

Non-covalent Bruton’s tyrosine kinase inhibitors (ncBTKi) are being investigated for the treatment of B-cell malignancies, including chronic lymphocytic leukemia (CLL). These agents hold promise for the treatment of CLL, especially CLL requiring treatment after prior discontinuation of covalent Bruton’s tyrosine kinase inhibitors such as ibrutinib, acalabrutinib and zanubrutinib for either intolerance or progression of disease. This review outlines current preclinical and clinical data for the use of ncBTKi in CLL. Recently reported data has shown durable responses for the ncBTKi pirtobrutinib in CLL patients previously treated with a covalent BTKi with a median PFS of 19.6 months. We also discuss recently discovered mechanisms of resistance to ncBTKis, including acquired mutations in the BTK protein not in the C481 position. In addition, we highlight ongoing clinical trials that are incorporating ncBTKis in CLL as monotherapy or in combination therapies with other agents. The results of these trials as well as ongoing research regarding mechanisms of resistance to ncBTKi and covalent Bruton’s Tyrosine Kinase inhibitors will be important to determine where ncBTKi may fit into the treatment of CLL.

**Abstract:**

Covalent Bruton’s tyrosine kinase inhibitors (cBTKi) have led to a paradigm shift in the treatment of chronic lymphocytic leukemia (CLL). These targeted oral therapies are administered as standard treatments in both the front-line and relapsed and/or refractory settings. Given their administration as a continuous therapy with a “treat-to-progression” strategy, limitations of their use include discontinuation due to toxicity or from progression of the disease. Non-covalent Bruton’s tyrosine kinase inhibitors (ncBTKi) distinguish themselves by binding reversibly to the BTK target, which may address the limitations of toxicity and acquired resistance seen with cBTKi. Several ncBTKis have been studied preclinically and in clinical trials, including pirtobrutinib and nemtabrutinib. Pirtobrutinib, which is now FDA approved for relapsed and/or refractory mantle cell lymphoma (MCL), has shown outstanding safety and preliminary efficacy in CLL in phase 1 and 2 clinical trials, with phase 3 trials underway. This agent may fill an unmet medical need for CLL patients requiring treatment after a cBTKi. Pirtobrutinib is particularly promising for the treatment of “double exposed” CLL, defined as CLL requiring treatment after both a cBTKi and venetoclax. Some patients have now developedacquired resistance to pirtobrutinib, and resistance mechanisms (including novel acquired mutations in BTK outside of the C481 position) have been recently described. Further study regarding the mechanisms of resistance to pirtobrutinib in patients without prior cBTKi exposure, as well as the potential for cross-resistance between cBTKi and ncBTKis, may be important to help inform where ncBTKis will ultimately fit in the treatment sequencing paradigm for CLL. An emerging clinical challenge is the treatment of CLL after ncBTKi discontinuation. Novel therapeutic strategies are being investigated to address the treatment of patients following disease progression on ncBTKis. Such strategies include novel agents (BTK degraders, bispecific antibody therapy, CAR T-cell therapy, PKC-beta inhibitors) as well as combination approaches incorporating a ncBTKi (e.g., pirtobrutinib and venetoclax) that may help overcome this acquired resistance.

## 1. Background: BTK Inhibitors in CLL

The covalent Bruton’s tyrosine kinase inhibitors (cBTKi) ibrutinib, acalabrutinib and zanubrutinib are now widely utilized standard treatment options for both the initial treatment of CLL and relapsed and/or refractory chronic lymphocytic leukemia (CLL) [1,2,3,4,5,6,7]. Multiple randomized clinical trials have shown an improved progression-free survival (PFS) with cBTKi treatment when compared to chemoimmunotherapy control arms [2,3,5,8]. This excellent clinical activity, including in patients with high-risk cytogenetic and molecular features, has led to a paradigm shift away from chemoimmunotherapy regimens towards targeted agent-based treatment programs [1,2,6,8,9,10]. Ibrutinib and acalabrutinib bind to the BTK protein irreversibly and decrease phosphorylation of BTKand therefore decrease downstream B-cell receptor signaling [4,11,12]. The B-cell receptor signaling pathway plays a critical role in the survival of CLL cells [4,11]. Importantly, as historically studied and currently approved for use as the standard of care, cBTKi are small-molecule inhibitors that are orally administered with the intent of continuous therapy until the disease progresses or until discontinuation is required due to unacceptable toxicity [2,6,9].

Despite outstanding clinical outcomes seen with ibrutinib, a subset of patients must discontinue ibrutinib due to toxicity or due to disease progression [13]. Retrospective data has shown that 41% of patients discontinued ibrutinib (median follow-up 17 months, with a total of 616 ibrutinib-treated patients included in study) [14]. Toxicity represented over half of all discontinuations. In the relapsed setting, atrial fibrillation (12.3%), infection (10.7%), pneumonitis (9.9%) and bleeding (9%) were the most common toxicities leading to discontinuation [14]. Discontinuations due to toxicity have also been more common than discontinuations due to disease progression in both additional retrospective series and clinical trials [1,6,15]. Acalabrutinib is a more selective inhibitor of BTK, with fewer off-target kinase effects [16]. Results from the randomized phase III ELEVATE-RR trial comparing ibrutinib to acalabrutinib for the treatment of relapsed and/or refractory CLL demonstrate the noninferior efficacy of acalabrutinib (HR 1.0) compared to ibrutinib and an improved safety profile with regards to including lower rates of all grade atrial fibrillation, diarrhea, arthralgia, and hypertension [17]. Zanubrutinib is a cBTKi that has been shown to have superior PFS to ibrutinib in the relapsed/refractory setting, with improved safety and fewer overall discontinuations of therapy due to AEs [7].

However, while acalabrutinib and zanubrutinib have addressed some toxicity concerns withthe cBTKi ibrutinib, therapy discontinuation due to disease progression and acquired resistance remains a limitation of the cBTKi class [12,18]. Mutations in the BTK protein at the binding site of ibrutinib at the C481 position have been detected in a majority of CLL patients at the time of clinical disease progression on the covalent BTKi ibrutinib [18,19,20]. Functionally, BTK C481 mutations have been shown to prevent the irreversible binding of ibrutinib to BTK, diminishing the ability of ibrutinib to block both BTK autophosphorylation and downstream B-cell receptor signaling [12]. In addition, a smaller number of patients have acquired mutations downstream of BTK in PLCG2, which have also been shown to confer resistance to ibrutinib [12,18]. BTK C481 and PLCG2 mutations have also been reported to occur in patients with disease progression during treatment with acalabrutinib [21]. Recently, data regarding resistance mechanisms for patients treated with zanubrutinib were reported. In a cohort of 13 zanubrutinib-treated patients, ten patients with progressive CLL had C481 mutations and seven patients had BTK L528W mutations, suggesting that acquired L528W mutations may play a role in zanubrutinib resistance in addition to C481 mutations [22].

There is no conclusive understanding as to what causes the development of resistance mutations to cBTKi. Current research has hypothesized that resistance mutations can arise in patients from evolutionary development of a BTK or PLCG2 mutant clone that allows an escape from BTK inhibition. In addition, it has been proposed that resistance arises from the competitive selection or outgrowth of drug resistant mutant clones already present at treatment initiation [18,19]. In general, more studies are needed to fully define the pathophysiologic development of cBTKi resistance.

Non-covalent BTK inhibitors (ncBTKi) reversibly bind to BTK, and hold promise to address some of the limitations of toxicity and resistance posed by cBTKi [23,24]. Here we review preclinical data for ncBTKi in CLL and review ongoing clinical trials of ncBTKi in patients with CLL. We also review recent research detailing mechanisms of resistance to non-covalent BTK inhibitors for patients with CLL. Finally, we preview existing data and potential options for the treatment of relapsed CLL following treatment with a ncBTKi.

## 2. Non-Covalent BTKis: Pre-Clinical and Clinical Data

As described above, cBTKis directly bind to the cysteine at residue 481 of BTK in the ATP binding pocket, effectively blocking the autophosphorylation of BTK tyrosine residue 223 (Y223). In contrast, ncBTKis interact with BTK using hydrogen bonds, ionic bonds and hydrophobic interactions [25]. This mechanism does not require C481, while allowing non-covalent BTKi to overcome the resistance that occurs from the BTK C481 point mutations that emerge in patients as a mechanism of resistance to cBTKi [26,27]. 

Despite the differences in the methods of inhibitor binding, structural studies of BTK with both cBTKi and ncBTKi show similarities in their required interactions with residues around 410, 430, 480, and 540, in addition to inhibitor-specific interactions. The ncBTKis (~26–29) have a slightly higher number of interactions than cBTKis (~19–22), possibly due to the size of the molecule, and due to the fact that more interactions are required for these inhibitors to achieve the affinity necessary for a reversible drug [28]. The weaker, non-covalent, reversible binding of these inhibitors to BTK suggests that the toxicity profiles of these inhibitors may be more favorable compared to a cBTKi [29,30,31].

There are several ncBTKis in development. These drugs have the potential to transform the treatment of B-cell malignancies, including CLL, and fill an unmet medical need for therapies in patients who are resistant or intolerant to treatment with cBTKis. In this review, we discuss ncBTKis, including vecabrutinib, fenebrutinib, nemtabrutinib, and pirtobrutinib, as well as discussing what is known regarding resistance mechanisms for ncBTKis.

### 2.1. Vecabrutinib 

Vecabrutinib, formally known as SNS-062, was among the first reversible BTKis to show potency against BTK with a half-maximal inhibitory concentration (IC^50^ 3nM). In addition to BTK, vecabrutinib also targets BLK, IGF1R, ITK, LCK, and TEC (all with an IC^50^ < 100 nM) but not epidermal growth factor receptor (EGFR), thereby reducing EGFR toxicities seen previously in covalent BTKi [26,32]. Vecabrutinib reversibly binds BTK [26]. A phase 1b study of vecabrutinib in patients with B-cell malignancies (NCT03037645), including CLL was conducted. The study had 39 patients (77% CLL) being treated with a dose escalation method until either disease progression or toxicity. All patients on the trial had received previous BTKi treatment; in addition, 45% had a BTK C481 mutation and 18% had a PLCG2 mutation [32,33]. Common treatment related AEs seen in patients included anemia (31%), headache (21%), dyspnea (21%), and fatigue (21%) [33]. The clinical development for this drug has since been terminated for B-cell malignancies [32,33].

### 2.2. Fenebrutinib

Fenebrutinib, formally known as GDC-0853, is another reversible ncBTKi that hydrogen binds to the K430, M477, and D539 residues of BTK [34,35,36]. Fenebrutinib targets BTK (IC^50^ 2.3 nM) with a minimal number of off-target effects and a low disassociation rate from BTK, making the extent of treatment comparable to irreversible covalent BTKi [25]. The first-in-human phase 1 trial for fenebrutinib (NCT01991184) included 24 patients, with 25% of patients having a BTK C481S mutation. Response rates for this inhibitor were the best in patients with CLL (50%, n = 7), with the overall response rate (ORR) for the trial being 33% [27,37]. Reported AEs for fenebrutinib included fatigue (37.5% any grade, 4.2% grade 3 or higher), nausea (33% any grade, 0% grade 3 or higher), diarrhea (29% any grade, 0% grade 3 or higher), thrombocytopenia (25% any grade, 8.3% grade 3 or higher), headache (21% any grade, 0% grade 3 or higher), dizziness (17% any grade, 0% grade 3 or higher), abdominal pain (17% any grade, 4.2% grade 3 or higher), and cough (17% any grade, 0% grade 3 or higher) [26,27,37]. Like vecabrutinib, fenebrutinib has also been discontinued in B-cell malignancies, but it still being tested in alternative diseases, such as multiple sclerosis (NCT04544449) [38].

### 2.3. Nemtabrutinib

Another reversible cBTKi is nemtabrutinib (formerly known as MK1026 and ARQ-531). Nemtabrutinib can maintain inhibitory pressure on the B-cell receptor (BCR) pathway in both wildtype and C481 mutant BTK [39]. It achieves this through the formation of hydrogen bonds to the E475 and Y476 residues of BTK [26,29,39]. Nemtabrutinib is less selective than other ncBTKi. It inhibits BTK (IC^50^ 0.85 nM), as well as SRC, AKT, and ERK, and has shown anti-proliferative activity in vitro against numerous hematological malignancies [39]. The phase 1 trial has shown its efficacy in CLL patients with both BTK mutations and activating PLCG2 mutations, as well as in patients with Richter Transformation (NCT03162436) [24,40]. This is likely due to the off-target inhibition of Lyn and Syk, two kinases shown to directly activate PLCG2 mutations. 

Data from the Bellwave-001 phase I/II study of nemtabrutinib in relapsed/refractory B-cell malignancies demonstrated an overall response rate of 56% (CR: 2, PR:15, PR-L: 15, n = 32 of 57 R/R CLL patients, median follow-up 8.1 months) [41]. The median duration of response was 24.4 months, with a median PFS of 26.3 months among responders. Notably, 95% of CLL patients in this study had prior cBTKi exposure and 42% of patients had prior cBTKi and BCL2 inhibitor treatment. The CLL study population was also high-risk from a cytogenetic and molecular perspective: 63% exhibited BTKC481S mutation, 32% exhibited TP53 mutation, 33% haddel17p, and 53% exhibited unmutated IGHV. In 112 patients with B-cell malignancies treated at the 65 mg daily dose of nemtabrutinib, the most common AEs of any grade were: dysgeusia (21%), decreased neutrophils (20%), fatigue (13%), nausea (12%), decreased platelets (12%), diarrhea (10%), and hypertension (10%). Forty percent (45 patients) had grade 3 or 4 treatment-related AEs (decreased neutrophils most common). The discontinuation rate due to AEs was 13% (n = 15 of 112 patients) [41].

### 2.4. Pirtobrutinib

Pirtobrutinib, formally known as LOXO-305, is a reversible, orally available, highly selective nc BTKi. Similar to other BTKis, pirtobrutinib works by blocking the ATP binding site on BTK; however, pirtobrutinib achieves greater than 300-fold selectivity for BTK in 98% (363/370) of kinases tested with an IC^50^ of 3.15 nM [42,43].

The Phase I/II BRUIN study (NCT03740529) is a phase I/II dose escalation and expansion trial of pirtobrutinib in patients with relapsed and/or refractory B-cell malignancies, which includes patients with CLL/SLL, mantle cell lymphoma (MCL), diffuse large B-cell lymphoma, Waldenström macroglobulinemia, follicular lymphoma, marginal zone lymphoma, Richter transformation, B-prolymphocytic leukemia, hairy cell leukemia, and primary CNS lymphoma, among other transformations (n = 773 patients) [44]. The study included 317 patients with relapsed and/or refractory CLL/SLL, including 282 previously treated with a prior cBTKi, of which 247 were evaluable for efficacy. The median number of prior therapies was three, with 41% of the cohort having received both a BCL2i and cBTKi previously; in addition, 79% of patients had received prior chemotherapy. Of patients with cytogenetic and molecular data available pretreatment, 38% had a BTK C481 mutation, 8% had a PLCG2 mutation, 39% for TP53 mutation, 29% for del17p, 85% for IGHV unmutated, and 42% had a complex karyotype. Reasons for discontinuation of the prior cBTKi included progressive disease (77%) and toxicity (23%) [44].

The overall response rate (ORR) for patients with CLL/SLL that had been treated with a prior cBTKi was 82.2% (CR: 4, PR: 177, PR-L: 22, SD: 26, n = 247); ORR was 79.0% for patients treated with a prior cBTKi and BCL2i (n = 100 patients) [44]. At a median follow-up of 19.4 months, the median PFS for the cohort of 247 patients was 19.6 months (95% CI: 16.9–22.1). For the “double exposed” patient cohort (i.e., patients with prior treatment with a cBTKi and BCL2i), the median PFS was 16.8 months (95% CI 13.2–18.7 months, median follow-up 18.2 months). Subgroup analyses revealed similar response rates across most subgroups when stratified using prior therapies and cytogenetic and molecular features, with the exception of PLCG2 mutated patients (ORR 55.6% vs. 83.9% for wildtype patients); however, there were a small number of patients with PLCG2 mutations, which may limit this result (n = 10 total). Notably, median PFS was not significantly different between patients with and without a baseline BTKC481 mutation [44].

Pirtobrutinib was well tolerated in the phase I/II BRUIN study. In the total study population of patients with relapsed and/or refractory B-cell malignancies, 2.6% of patients (n = 20 of 773 patients, median time on treatment for cohort 9.6 months) had discontinued treatment due to an adverse event, which was notably lower than historical discontinuation with cBTKis reported in real world studies and clinical trials [1,6,9,14,44]. The five most common any grade AEs to emerge were fatigue (28.7%), diarrhea (24.2%), neutropenia (24.2%), contusion (19.4%), and cough (17.5%). The most common grade 3 or higher AE was neutropenia (20.4% of patients), followed by anemia (8.8% of patients). Any grade hypertension was present in 9.2% of patients, with atrial fibrillation/flutter in 2.8% of patients. Grade 3 or higher hemorrhage was present in 1.8% of patients.

Pirtobrutinib is now being further explored in four global, randomized phase III trials in patients with CLL/SLL, which has been outlined in Table 1. Of note, based on the results of patients with MCL treated with pirtobrutinib on the phase I/II BRUIN study, pirtobrutinib was granted accelerated approval by the Food and Drug Administration for the treatment of relapsed and/or refractory mantle cell lymphoma (at least two prior lines of therapy, including a prior cBTKi) in January 2023 [45].

## 3. Mechanisms of Resistance to Non-Covalent BTK Inhibitors

While the previously described ncBTKi can inhibit both the wildtype and the cBTKi-resistant C481 mutant BTK, patients on these early-phase clinical trials have discontinued treatment due to the progression of their disease, demonstrating that acquired resistance to treatment is a limitation of their use [46,47]. Despite outstanding outcomes compared to historical controls (median PFS of 19.6 months at a median follow-up of 19.4 months), patients have experienced progression of CLL on pirtobrutinib [44].

Recent research has demonstrated that mutations in BTK outside of the C481 position can confer resistance to the ncBTKi pirtobrutinib [47]. Using peripheral blood, bone marrow and lymph node samples taken pretreatment, on-treatment, and at the time of clinical disease progression from CLL patients treated during the BRUIN study at a single center, the authors performed next generation sequencing and identified novel mutations at the time of clinical disease progression while taking pirtobrutinib. These mutations clustered within the tyrosine catalytic domain of BTK. The paper included nine patients with CLL who had been treated with pirtobrutinib and discontinued treatment due to the progression of their disease (the best overall response rate to pirtobrutinib for these patients was 44%). In three of the nine patients, progression of disease manifested as biopsy-proven Richter’s transformation to diffuse large B-cell lymphoma. There were seven patients with acquired BTK mutations. While all seven patients had previously been treated with ibrutinib and discontinued ibrutinib due to the progression of CLL, newly acquired BTK mutations occurred in patients with and without baseline BTK C481 mutations; in two of four the patients with pre-existing BTK C481 mutations, the BTK C481 mutation was suppressed during treatment with pirtobrutinib. In addition, three patients had pre-existing mutations in PLCG2, which persisted at the time of clinical disease progression while taking pirtobrutinib [47].

Newly identified BTK mutations that conferred resistance to pirtobrutinib include L528W, V416L, A428D, M437R, and T474I. To functionally characterize these mutations, the authors expressed mutant constructs into BTK dependent lymphoma cell lines. These mutants all contained bulky sidechain point substitutions that clustered in the catalytic domain of BTK and appeared to physically impede drug binding within the ATP binding pocket [47,48].

Given the similar binding moieties of BTK inhibitors, this data also showed that non-C481 BTK mutants (V416L, A428D, M437R, T474I, and L528W) confer resistance not only to pirtobrutinib, but to other covalent (ibrutinib) and non-covalent (nemtabrutinib, vecabrutinib, and fenebrutinib) BTKis (Figure 1) [47]. Intriguingly, the BTK mutants V416L, A428D, M437R, and L528W showed decreased kinase activity in BTK, based on the absence of BTK Y223 phosphorylation [47,48,49]. Despite diminished BTK activation, authors showed sustained downstream activation of AKT, ERK, and hyperactivated calcium release even in the presence of the BTKi [47].

In a recent study, Qi et al. generated BTKi resistant Rec-1 cell lines through long-term exposure to covalent (ibrutinib) and non-covalent (pirtobrutinib, vecabrutinib, nemtabrutinib, and fenebrutinib) BTKi [48]. They identified previously described patient-derived BTK resistance mutants C481F, V416L, A428D, and L528W, as well as novel mutations: L528S, G409R, G480R, and D539H. Similar to results reported by Wang et al. [47], they found that these mutations cluster in the kinase domain of BTK and negatively affected kinase activity [48]. Interestingly, they found no BTK or PLCG2 mutations in the nemtabrutinib resistant cells, alluding to an alternative signaling mechanism of resistance. They found an increased expression of insulinlike growth factor receptor (IGF1R) and increased sensitivity to IGF1R inhibitors in all BTKi resistant cell likes, except G409R [48].

Another group studying resistance to ncBTKi performed a whole exome sequencing of samples from two patients with disease progression while taking pirtobrutinib, so as to provide evidence of resistance through multiclonal alternative site BTK mutations [50]. In the first patient studied, they found that gatekeeper mutations identified in patients, such as T474I, showed strong clonal selection during exposure to pirtobrutinib. After further studies, the authors identified another BTK mutant, M477I, within the same clone (2% cancer cell fraction) and a subclone carrying acquired mutations in RAC1 and TP53BP2. The second studied patient harbored both the L528W and C481R, another kinase dead mutant, at similar VAF at the time of resistance. A third clone identified in that patient showed a C481S mutant. While this mutant is known to be sensitive to pirtobrutinib, the authors suggested that resistance could be driven by the presence of the TP53 R196 mutant. In line with previously described studies, and despite the predominance of the inactivating mutations L528W and C481R, the authors also observed intact distal BCR signaling [50].

While the exact signaling mechanism of resistance in these inactivating ncBTKi mutations requires further study, recent articles have reported a noncatalytic scaffolding function of BTK with HCK or LYN or dependencies of kinase-deficient BTK mutants on TLR9, UNC93B1, CNPY3 [48,49,50,51]. Additionally, as seen in cBTKi resistance, alternative receptor tyrosine kinase (RTK) signaling through such pathways as the RAS-MAP kinase, NFkB, or PI3K-mTOR can also contribute to resistance through complementary survival signaling [46,52,53,54,55].

Given the clinical promise of ncBTKi, the recent identification of acquired resistance mutations to ncBTKi that negatively affect the kinase activity of BTK (rendering it kinase deficient/dead) highlights a crucial unmet need for understanding the noncatalytic function of BTK in BCR signaling. Additional studies are also required to functionally characterize those gain-of-function PLCG2 mutants (S707F, E1139del, D1140E and D1144G) identified from ncBTKi resistance (Figure 1) [46,47,48].

## 4. ncBTKi in CLL: Where Will These Agents Fit into the Treatment Paradigm?

While pirtobrutinib was recently granted accelerated approval by the FDA for patients with MCL, pirtobrutinib and other ncBTKis remain investigational for patients with CLL. As a monotherapy, pirtobrutinib holds promise to fill an unmet need for patients with not only progression after cBTKi treatment, but also for patients needing CLL therapy after treatment with both a cBTKi and venetoclax. This patient population of “double exposed” CLL patients who have been treated previously with both classes of targeted agents marks an area of great unmet need for patients with CLL [56,57,58,59]. This may include patients who discontinue treatment with targeted therapies due to disease progression or from intolerance due to its side effects. Notably, while CIT and phosphatidylinositol-3-kinase (PI3K) inhibitors are the standard treatment options for this patient population, the studies leading to their approvals did not include patients treated with prior cBTKi and venetoclax [60,61]. Further, retrospective analyses suggest that responses for subsequent standard therapies are not durable following cBTKi and venetoclax exposure [57,58,62]. In the recently reported updated BRUIN study results, the median PFS for pirtobrutinib was 16.8 months for patients treated with both prior BTKi and BCL2i (median follow-up 18.2 months; median number of prior therapies five) [44]. Pirtobrutinib monotherapy may have a role in this patient population in the future, given these promising early clinical outcomes. There are ongoing randomized trials of pirtobrutinib vs. PI3Ki and CIT (Table 1, BRUIN-CLL-313 and BRUIN-CLL-321).

Pirtobrutinib has also been studied in combination with other agents, including venetoclax and rituximab. The phase Ib portion of the BRUIN study evaluated the safety of pirtobrutinib + venetoclax, and pirtobrutinib + venetoclax + rituximab in patients with relapsed/refractory CLL [63]. The combination was overall well tolerated, with no dose-limiting toxicities, with the most common treatment emergent adverse events being a neutrophil count decrease, at 36%; nausea was present in 32% of the sample, fatigue in 32%, diarrhea in 28%, and constipation in 24%. The overall response rate was 95.5% (with a median follow-up of 9 months). Pirtobrutinib is now being further explored in combinations, including in a global, randomized phase III trial (pirtobrutinib + venetoclax + rituximab, versus venetoclax + rituximab, see Table 1). In addition, there are other ongoing studies looking at pirtobrutinib and venetoclax in previously untreated CLL (NCT05677919), as well as pirtobrutinib, venetoclax, and obinutuzumab, likewise in previously untreated CLL (NCT05536349).

Whether pirtobrutinib will be moved into earlier lines of therapy, including front-line therapy or prior to the use of cBTKi, remains an unanswered question. There is an ongoing trial of pirtobrutinib versus ibrutinib in previously untreated CLL (NCT05254743, Table 1) which will provide important efficacy and safety data on pirtobrutinib in a front-line setting. While randomized clinical trial data will be vital to answering whether ncBTKi may have a role in earlier lines of therapy, the mechanisms of resistance and potential for cross-resistance will likely play an important role in whether ncBTKi should precede cBTKi, especially in a chronic, incurable disease such as CLL. For example, preclinical work has shown that acquired mutations (such as BTKL528W) seen at the time of clinical resistance to pirtobrutinib also confer resistance to ibrutinib [47]. This potentially suggests that ibrutinib, and perhaps other cBTKi, may have a limited role in some patients with resistance to pirtobrutinib. However, it should be noted that all patients with acquired BTKL528W mutations at the time of clinical resistance to pirtobrutinib received prior treatment with ibrutinib, and it is unknown if these mutations will arise in patients treated with pirtobrutinib as their first BTKi [47]. Prior to informing clinical management, further research is needed regarding mechanisms of resistance to the different cBTKis using consistent sequencing methods, as well as whether clinically significant cross-resistance due to acquired mutations will inform sequencing or selection between cBTKis and ncBTKis. Further clinical trials and studies of resistance and sequencing are needed to study the optimal sequence of cBTKi and ncBTKi, as well as their mechanisms of resistance and cross-resistance.

## 5. Treatment of CLL after ncBTKi: An Emerging Challenge

An emerging challenge will be treatment of CLL after ncBTKi discontinuation. Currently, there is limited data available on subsequent therapies. A retrospective analysis of patients with CLL and Richter’s transformation (RT) found that for CLL patients who are venetoclax-naïve, venetoclax has excellent clinical activity following ncBTKi discontinuation (with a median PFS of 14 months and a median follow-up of 10.5 months, across eight patients). Standard therapies (including CIT and PI3Ki) had low overall response rates and responses were not durable, while patients treated with CAR T-cell therapy and allogeneic stem cell transplant had higher ORRs [13].

There are ongoing trials of several different classes of agents that may hold promise after ncBTKi discontinuation. These novel strategies include, but are not limited to, BTK degraders (NX2127, NCT04830137 [64]; BGB-16673, NCT05006716), bispecific antibody therapy (mosunetuzumab, NCT05091424; epcoritamab, NCT04623541), and PKC-beta inhibitors (MS-553, NCT03492125) [65]. In addition, given the clinical efficacy demonstrated in the TRANSCEND CLL study, CD19-directed CAR T-cell therapy should also be explored further in this patient population [66].

## 6. Conclusions

Therapies targeting BTK continue to revolutionize the treatment of CLL. While covalent inhibitors of BTK (such as ibrutinib, acalabrutinib, and zanubrutinib) have improved patient outcomes, their use may be limited by toxicity as well as by resistance. The non-covalent BTKi pirtobrutinib has demonstrated outstanding early tolerability and clinical activity, including durable responses in relapsed and/or refractory patients previously exposed to covalent BTKi or BCL2i. While these ncBTKi show promising efficacy in CLL, a subset of patients has developed resistance due to BTK and PLCG2 alterations. Long-term safety and efficacy data remain unknown. An emerging clinical challenge will be the treatment of patients after acquiring resistance to ncBTKi. Longer follow-ups from the early clinical trials of ncBTKis, as well as randomized studies of ncBTKis versus current standards of care, will help inform where these agents will fit into the CLL sequencing paradigm. More research is required regarding the mechanisms of acquired resistance to cBTKis and ncBTKis, as well as the potential for cross-resistance between cBTKi and ncBTKi. In addition, preclinical and clinical studies of novel combination approaches and novel therapeutic strategies such as BTK degraders, bispecific antibodies and PKC-beta inhibitors will be essential for improving patient outcomes.

## Figures and Tables

**Figure 1 cancers-15-03648-f001:**
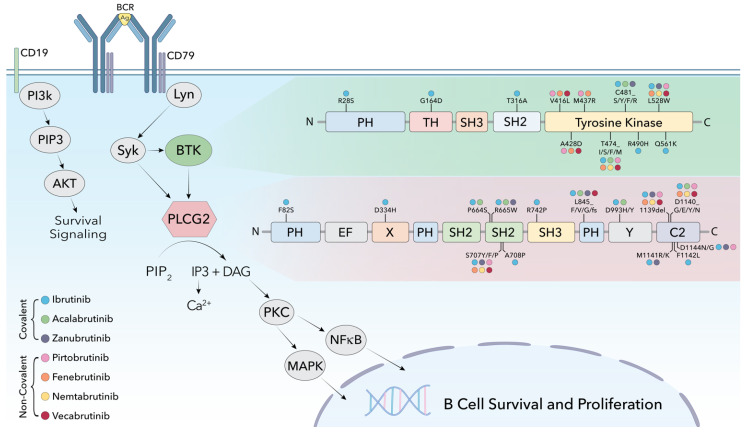
B-cell receptor mediated signaling, highlighting BTK and PLCG2 kinases. Resistance causing mutations are mapped on to domains of BTK and PLCG2, showing their location and corresponding covalent (ibrutinib, acalabrutinib and zanubrutinib) and non-covalent (pirtobrutinib, fenebrutinib, nemtabrutinib and vecabrutinib) inhibitor resistances.

**Table 1 cancers-15-03648-t001:** Additional Studies of Pirtobrutinib in CLL/SLL.

Study	Study Description	Study Population	Primary Outcome
BRUIN-CLL-313 (NCT05023980)	Pirtobrutinib vs. Bendamustine + Rituximab	Previously untreated CLL/SLL	PFS
BRUIN-CLL-314 (NCT05254743)	Pirtobrutinib vs. ibrutinib	Previously untreated CLL/SLL	ORR
BRUIN-CLL-321 (NCT04666038)	Pirtobrutinib vs. investigator’s choice (idelalisib + rituximab vs. bendamustine + rituximab)	R/R CLL/SLL	PFS
BRUIN-CLL-322 (NCT04965493)	Pirtobrutinib + venetoclax + rituximab vs. venetoclax + rituximab	R/R CLL	PFS
MIRACLE (NCT05677919)	Pirtobrutinib + Venetoclax	Previously untreated CLL	Undetectable minimal residual disease (<1/10^4^) after 15 cycles of treatment
Time-limited triplet combination of pirtobrutinib, venetoclax and obinutuzumab (NCT05536349)	Pirtobrutinib + Venetoclax + Obinutuzumab	Previously untreated CLL or Richter Transformation (RT)	Undetectable measurable residual disease rate of PVO in patients with CLL cohort; ORR for RT cohort

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
