# Peer review of "Non-Covalent Bruton’s Tyrosine Kinase Inhibitors in the Treatment of Chronic Lymphocytic Leukemia"

_cancers, 2023, doi:10.3390/cancers15143648_

Round 1

Reviewer 1 Report

Interesting paper.  Only minor comments.

1) Line 25-26: repetition in the same sentence?

«Pirtobrutinib is particularly promising for the treatment of “double exposed” CLL, or CLL requiring treatment after both a cBTKi and venetoclax»

2) Line 48-49

What about acalabrutinib? 

3) Line 169. Please clarify :

…pirtobrutinib achieves greater than 300-fold selectivity 168 for BTK vs 363/370 (98%) of other kinases with an IC50 of 3.15n 

4) Line 202

Please revise:

The most common grade  3 or higher AE was neutropenia (20.45 of patients) 

5) It could be interesting to more clearly explains the physiopathologic hypothesis leading to development of covalent BTKi resistance.

Author Response

Reviewer 1 Comments:
Thank you for your thoughtful review. Please find our responses to the comments below.

Interesting paper.  Only minor comments. 

1) Line 25-26: repetition in the same sentence? 

«Pirtobrutinib is particularly promising for the treatment of “double exposed” CLL, or CLL requiring treatment after both a cBTKi and venetoclax» 

Thank you for the opportunity to clarify this sentence. We have revised this sentence to read “pirtobrutinib is particularly promising for the treatment of “double exposed” CLL, defined as CLL requiring treatment after both a cBTKi and venetoclax.” This should clarify the definition of double exposed CLL for the reader.

2) Line 48-49 

What about acalabrutinib?  

Lines 48-49 read “Ibrutinib and acalabrutinib bind to the BTK protein irreversibly and decrease phosphorylation of BTK and therefore downstream B-cell receptor signaling.” We invite the author to provide additional information on what should be clarified in lines 48 and 49.

3) Line 169. Please clarify :

…pirtobrutinib achieves greater than 300-fold selectivity 168 for BTK vs 363/370 (98%) of other kinases with an IC50 of 3.15n 

Thank you for the opportunity to clarify this sentence. We have revised the text to read: Similar to other BTKis, pirtobrutinib works by blocking the ATP binding site on BTK; however, pirtobrutinib achieves greater than 300-fold selectivity for BTK in 98% (363/370) of kinases tested with an IC50 of 3.15nM [42,43].”

4) Line 202

Please revise:

The most common grade  3 or higher AE was neutropenia (20.45 of patients) 

We have revised this to read “20% of patients.”

5) It could be interesting to more clearly explains the physiopathologic hypothesis leading to development of covalent BTKi resistance.

Thank you for this comment. We have added the following text on page 2 to address this comment: “There is no conclusive understanding as to what causes the development of resistance mutations to cBTKi. Current literature hypothesizes that resistance mutations can arise in patients from evolutionary development of a BTK or PLCG2 mutant clone that allows escape from BTK inhibition. In addition, it has been proposed that resistance arises from the competitive selection or outgrowth of drug resistant mutant clones already present at treatment initiation [18,19]. In general, more studies are needed to fully define the pathophysiologic development of cBTKi resistance.”

Reviewer 2 Report

The revue on Non-Covalent Bruton's Tyrosine Kinase Inhibitors in the Treatment of CLL is well written. The paper addresses the correct issues in an understandable and effective way. Correct the distribution and order in which subchapters are displayed.

Minor

On page 7, line 295, one could speak more clearly of “intolerant” patients.

On page 8 line 323 the ongoing trial of the comparison between Pirtobrutinib vs Ibrutinib in order to understand if the non-covalent inhibitor can be introduced at the forefront of CLL treatment is correctly mentioned. However, currently ibrutinib is practically no longer used as monotherapy in the US and Europe is moving in the same direction. Perhaps a comment could be introduced by the Authors in the discussion highlighting that the absence of a comparison with Acalabrutinib and Zanubrutinib could represent a byass especially as regards AEs.

Author Response

Reviewer 2 Comments:

Thank you for your thoughtful review. Please find our response to your comments below:

The revue on Non-Covalent Bruton's Tyrosine Kinase Inhibitors in the Treatment of CLL is well written. The paper addresses the correct issues in an understandable and effective way. Correct the distribution and order in which subchapters are displayed.

Thank you for this comment. We have reviewed the order of the paper and the display of different sections for this review. We invite the reviewer and journal editor to clarify what is requested in terms of changing the order in which subchapters are displayed.

Minor

On page 7, line 295, one could speak more clearly of “intolerant” patients.

We have added a sentence to incorporate that these patients may include both patients who discontinue targeted agents due to disease progression or intolerance. “This may include patients who discontinue treatment with targeted therapies due to disease progression or intolerance due to side effects.”

On page 8 line 323 the ongoing trial of the comparison between Pirtobrutinib vs Ibrutinib in order to understand if the non-covalent inhibitor can be introduced at the forefront of CLL treatment is correctly mentioned. However, currently ibrutinib is practically no longer used as monotherapy in the US and Europe is moving in the same direction. Perhaps a comment could be introduced by the Authors in the discussion highlighting that the absence of a comparison with Acalabrutinib and Zanubrutinib could represent a byass especially as regards AEs.

Thank you for this comment. The authors acknowledge that data of pirtobrutinib versus other covalent BTKis including acalabrutinib and zanubrutinib may potentially be valuable in the future. However, there are currently no ongoing trials that we are aware of comparing pirtobrutinib to second-generation covalent BTK inhibitors in the front-line setting. In addition, there is no data on pirtobrutinib monotherapy safety or efficacy that has been reported to date in the front-line setting and this trial will be important to establish the efficacy and safety in the front-line setting. Therefore, we added a clarification to the sentence to read:  There is an ongoing trial of pirtobrutinib versus ibrutinib in previously untreated CLL (NCT05254743, Table 1) which will provide important efficacy and safety data on pirtobrutinib in the front-line setting.”